# Synthesis, Structural, Morphological and Thermal Characterization of Five Different Silica-Polyethylene Glycol-Chlorogenic Acid Hybrid Materials

**DOI:** 10.3390/polym13101586

**Published:** 2021-05-14

**Authors:** Michelina Catauro, Pavel Šiler, Jiří Másilko, Roberta Risoluti, Stefano Vecchio Ciprioti

**Affiliations:** 1Department of Engineering, University of Campania “Luigi Vanvitelli”, Via Roma 29, I-81031 Aversa, Italy; 2Materials Research Centre, Faculty of Chemistry, Brno University of Technology, Purkyňova 118, CZ-61200 Brno, Czech Republic; siler@fch.vut.cz (P.Š.); masilko@fch.vut.cz (J.M.); 3Department of Chemistry, Sapienza University of Rome, Piazzale Aldo Moro 5, Building CU014, I-00185 Rome, Italy; roberta.risoluti@uniroma1.it; 4Department of Basic and Applied Science for Engineering (S.B.A.I.), Sapienza University of Rome, Via del Castro Laurenziano 7, Building RM017, I-00161 Rome, Italy

**Keywords:** sol–gel method, silica-based materials, polyethylene glycol, silica-PEG hybrids, chlorogenic acid

## Abstract

The present study investigated the structure, morphology, thermal behavior, and bacterial growth analysis of novel three-component hybrid materials synthesized by the sol-gel method. The inorganic silica matrix was weakly bonded to the network of two organic components: a well-known polymer such as polyethylene glycol (PEG, average molar mass of about 4000 g/mol), and an antioxidant constituted by chlorogenic acid (CGA). In particular, a first series was made by a 50 wt% PEG-based (CGA-free) silica hybrid along with two 50 wt% PEG-based hybrids containing 10 and 20 wt% of CGA (denoted as SP50, SP50C10 and SP50C20, respectively). A second series contained a fixed amount of CGA (20 wt%) in silica-based hybrids: one was the PEG-free material (SC20) and the other two contained 12 and 50 wt% of PEG, respectively (SP12C20 and SP50C20, respectively), being the latter already included in the first series. The X-ray diffraction (XRD) patterns and scanning electron microscope (SEM) images of freshly prepared materials confirmed that all the materials were amorphous and homogeneous regardless of the content of PEG or CGA. The thermogravimetric (TG) analysis revealed a higher water content was adsorbed into the two component hybrids (SP50 and SC20) because of the availability of a larger number of H-bonds to be formed with water with respect to those of silica/PEG/CGA (SPC), where silica matrix was involved in these bonds with both organic components. Conversely, the PEG-rich materials (SP50C10 and SP50C20, both with 50 wt% of the polymer) retained a lower content of water. Decomposition of PEG and CGA occurred in almost the same temperature interval regardless of the content of each organic component. The antibacterial properties of the SiO_2_/PEG/CGA hybrid materials were studied in pellets using either *Escherichia coli* and *Enterococcus faecalis*, respectively. Excellent antibacterial activity was found against both bacteria regardless of the amount of polymer in the hybrids.

## 1. Introduction

Human body cells are able to counteract under physiological free radicals conditions due to the presence of some enzymes able to manage redox homeostasis. However, when pro-oxidant species are overproduced and/or the biological repairing mechanisms are not so effective, the homeostatic processes may fail, thus compromising the usual pro-oxidative/antioxidative cellular balance. Under this severe condition, namely “oxidative stress”, reactive oxygen and nitrogen species (denoted as ROS and RNS, respectively), and consequently lipid peroxidation (LPO) products commonly promote tissue damage and cellular injury [1,2]. The study of the thermo-oxidative behavior of ultra-high molecular weight polyethylene (UHMWPE)-based nanocomposites recently revealed that sonication of carbon nanotubes has a detrimental effect on the thermo-oxidative stability of nanocomposites, especially for long exposure times [3].

In recent years, silica-based materials, with particular reference to organic-inorganic hybrids [4,5,6] and calcium bioactive glasses [7,8,9], have received increasing attention due to their relevant biological properties. In particular, they commonly show high biocompatibility, thus providing positive biological effects because their reaction products are able to promote cell–material interactions and cell invasion [10]. Furthermore, they are recognized to facilitate soft and hard tissue regeneration [10].

However, it is well known that it is possible to combine biopolymers and inorganic materials to develop highly innovative materials used for biomedical application. For example, biopolymers can be used to design different pharmaceutical forms and to encapsulate different types of nanosystem in the polymer matrix [11,12].

Biocompatibility of the silica-based materials may be improved by adding an appropriate defined amount of a polymer, such as polyethylene glycol (PEG) that increases the hydrophilicity of the obtained organic–inorganic hybrid materials and, therefore, improving cell adhesion and growth [13,14]. PEG is a biocompatible, versatile polymer, used for many biomedical purposes [13,14,15], including polymer-based materials for drug delivery [16,17]. The reduction of toxicity and the extension of the circulation time of many drug nanocarriers, were improved by the addition of PEG in the material [14,16,18]. In fact, SiO_2_/PEG hybrid materials have even been proposed as osteochondral regeneration matrices [19] and in drug delivery applications [19,20,21,22].

Chlorogenic acid (CGA), a non-flavonoid polyphenolic compound containing both aromatic and aliphatic groups, is an intermediate of secondary metabolic of plants obtained by esterification of hydroxycinnamates (such as caffeic acid) with quinic acid. It is commonly found in several plants and fruits, such as honeysuckle, Cortex Eucommiae, Semen Coffea Arabica, and green tea [23,24]. It has interesting biological and pharmaceutical properties that confer many benefits to human health. In fact, CGA can act to protect, prevent or attenuate the progression of certain skin disorders: minor diseases like wrinkles and acne or more serious one, such as cancer [25]. It also exerts protective action against induced oxidative stress, inflammation [26], promotes osteoblastogenesis in human adipose tissue-derived mesenchymal stem cells [27] or anti-inflammatory activity [26,28]. CGA also showed an antibacterial property [29] and enhanced the intrinsic cellular tolerance against oxidative insults either by activating survival/proliferation pathways or by increasing antioxidant capacity [30], thus making it an excellent candidate as an alternative to conventional antibacterial therapy [31,32]. Moreover, CGA is found in many dietary foods and it is widely bioavailable in humans because of the above-mentioned properties, thus exerting a beneficial action in the prevention and treatment of cancer [33,34]. However, its activities, as well as those of other polyphenolic compounds, are known to be limited to a short residence in the body [35].

In recent years, thermal analysis techniques and calorimetry have received increasing attention among scientists for their ability to characterize polymeric and composite materials for many kinds of possible applications. A large number of papers dealt with the thermal behavior study of different classes of organic, inorganic, and hybrid materials [36] as well as coordination compounds [37] with a view to assessing their thermal stability or to checking the eventual occurrence of fire accidents or evaporation of pollutants. Some investigations on environmentally hazardous pollutants, like herbicides and pesticides are worth noting since these substances may fall through the vapor phase directly on the soil surface or those washed onto the soil may be degraded or transported within the soil [38]. Therefore, these studies are of crucial importance to support theoretical models for prediction purposes [39,40].Furthermore, organomodified montmorillonite nanocomposites [41], plastic wastes [42,43] and light-cured resins [44] may undergo thermal degradation, combustion or pyrolysis (depending on the reaction temperature and the extent of oxygen consumption), and the results obtained may be used to draw relevant conclusions about a possible recycling procedure for energy recovery and/or to produce valuable chemical oily substances.

In common practice, the thermal behavior studies of materials are considered as a preliminary means of investigation aiming at detecting the most suitable temperature ranges at which pretreatment could be carried out to induce a given property for a relevant application [45,46]. In some cases, the objective is to remove water physically or chemically bond it to the inner structure of the materials [47], by heating the materials up to 90–120 °C for a defined time interval. In other cases it is necessary to reach higher temperature, where a transformation may occur that can change their characteristics remarkably, as for annealing of amorphous glass materials [48,49], to enhance the thermal stability of active components in drug formulations [50]. Alternatively, heat treatment is carried out with a view to improving the structural and thermal properties of metallic glasses [51], to induce upon heating in vacuo annealing and relaxation of amorphous solid water approaching the structure of hyperquenced glassy water [52]. Many materials and composites (antibiotics [53], polyhedral oligomeric silsesquioxane nanocomposites [54,55], polymers [56,57], organic–inorganic hybrids [58]) are often investigated by processing experimental thermogravimetric/differential scanning calorimetry (TG/DSC) data to obtain kinetic parameters related to thermal decomposition reactions aiming at determining parameters for assessing their thermal stabilities.

Papageorgiou and co-workers found that polymer crystallization may obey several different models, but the Prout–Tompkins one usually provides a more appropriate description of the process than that of Avrami [59]. As a consequence, an a priori assumption that the polymer crystallization kinetics follows the Avrami model may lead to determine erroneous crystallization rate constants if this assumption is not supported by experimental evidences.

In order to preserve and possibly enhance their bioactivity, in this study we aimed at producing hybrid materials by trapping different amount of CGA and/or PEG into a silica matrix in order to obtain biomaterials with new additional biological properties. The mutual effect of CGA on silica/PEG hybrids (containing 50 wt% of the polymer) and that of PEG on the silica/CGA materials (with 20 wt% of CGA), obtained through the sol-gel technique have been investigated, similarly to what has been recently made with a different composition [60]. Careful attention must be paid to the selection of the more appropriate synthesis method to prepare polymer or nanocomposite materials [61,62]. Recently, a simple and versatile method to prepare a multifunctional Dual Crosslinking (DC) single network polyethylene glycol polyurethane (PEG–PU) hydrogels with excellent mechanical properties, with promising developments in the biomedical field [63]. The sol-gel method uses a simple and versatile process, particularly suitable for preparing ceramics and glass materials at low temperatures. It is recognized to have some advantages with respect to other traditional methods [58]. Among them it is worth mentioning the purity of the products and the possibility of incorporating thermolabile molecules.

The growth of a Gram-negative and a Gram-positive bacteria (*Escherichia coli* and *Enterococcus faecalis* in this study) was monitored by increasing the PEG amount in these materials with a fixed content (20 wt%) of CGA. The corresponding results were discussed with a view to checking the role of the polymer in improving the intrinsic antibacterial properties.

## 2. Materials and Methods

Five novel three-component hybrid materials including silica (S), polyethylene glycol (PEG = P) and chlorogenic acid (CG = C) were synthesized at room temperature under atmospheric pressure according to the sol-gel method. The tetraethyl orthosilicate (TEOS, Si(OC_2_H_5_)_4_, Sigma-Aldrich, Steinheim am Albuch, Germany), was added dropwise to a 99.8% ethanol solution containing purchased by (Sigma-Aldrich, Steinheim am Albuch, Germany), distilled water and nitric acid HNO_3_ ≥ 65%, (Sigma-Aldrich, Steinheim am Albuch, Germany), used as a catalyst. After stirring each solution for 15 min equal rates of the resulting mixtures were added to different volumes of a solution containing a suitable content of polyethylene glycol (PEG 400), (Sigma-Aldrich, Steinheim am Albuch, Germany) with ethanol in order to have 50 wt.% of PEG in a common silica matrix [14,46]. 10 and 20 wt% of an antioxidant agent (CGA) were entrapped within the silica network of the 50 wt% PEG-based (SP50) hybrids according to Table 1. Another series of materials was prepared by adding 12 and 50 wt% of PEG to a 20 wt% CGA silica-based (SC20) hybrid material (Table 1), where SP50C20, which has the larger amount of both PEG and CGA, was included in both series.

The analyzed materials were finely ground, homogenized, and average bulk composition was determined. X-ray diffraction analysis was performed using an Empyrean (Panalytical) X-ray diffractometer with Bragg–Brentano parafocusing geometry, supplied by a PIXcel3D detector and using CuKα radiation (λ = 1.5418 Å). The experiments were performed within the values of 2θ ranging from 5 to 120° with an angular step of 2θ equal to 0.013° and 25 s duration, using automatic divergence slits to maintain the constant irradiation of the sample area. The measurements were repeated four times and then summed to enhance the signal to noise ratio.

The structural analysis of surfaces and element analysis was performed on a ZEISS scanning electron microscope (EVO LS10 model) supplied by an OXFORD X-Max 80 mm^2^ energy dispersive analyzer (New York, NY, USA). All measurement parameters are always shown at the bottom of all the figures.

In order to confirm the stoichiometry of the synthesized materials the Si content was evaluated by energy-dispersive X-ray spectroscopy (EDS) measurements, similarly to what has been recently made for Ca-containing silicate gel-glasses and limestone, respectively [7,64].

The nature of the bonds between the silica matrix and each of the organic component in the hybrid materials was evaluated by Fourier transform infrared (FTIR) spectroscopy in an effort to correlate it with the content both polymer and antioxidant compound. To this end, a Prestige 21 (Shimadzu, Japan) instrument was used, equipped with a DTGS KBr (Deuterated Tryglycine Sulphate with potassium bromide windows) detector, with a resolution of 4 cm^−1^ (60 scans). The spectra under the transmittance mode were recorded in the 400–4000 cm^−1^ range. Disks with a diameter of 13 mm, a thickness of 2 mm, containing 200 mg of KBr with 1 wt% of sample were obtained by pressing the sample powders into a cylindrical holder using a Specac manual hydraulic press. The Prestige software (IR solution) was used to analyze the FTIR spectra.

The thermal behavior of the hybrid materials was studied using a TGA7 apparatus (PerkinElmer, Waltham, MA, USA) equipped with platinum crucibles. Approximately 3–5 mg of samples (precisely weighed) was heated from ambient temperature to 1000 °C at a heating rate of 10 °C/min under argon atmosphere at 100 mL/min. Further details on calibration of temperature are reported elsewhere [65]. Following the equipment recommendations provided by the company, the calibration of temperature was performed using the Curie-point transition of very pure standard metals or alloys. In this study the endpoint temperatures of Ni and alumel at 163 and 354 °C, respectively, were considered for this purpose. Diagnostics and acquisition of the TG data were made using the Pyris software (Thermo Fisher Scientific Inc., Waltham, MA, USA) as ASCII files and processed by a V-JDSU Unscrambler Lite (Camo software AS), (Oslo, Norway).

The antibacterial properties of the SiO_2_/PEG/CGA hybrid materials were studied in pellets using *Escherichia coli* and *Enterococcus faecalis*, Gram-negative (ATCC 25922) and Gram-positive (ATTC 29212) bacteria, respectively. *Escherichia coli* was growth on TBX medium (Tryptone Bile X-Gluc), while *Enterococcus faecalis* was growth in Slanetz Bartley agar base both provided by Liofilchem (Roseto degli Abruzzi, TE, Italy). For diluting bacteria pellets autoclaved saline water (9 g of NaCl in 1 L) was provided. For the analysis of antibacterial properties, the following procedure was used: each sample was grounded to obtain a fine powder. Then, 100 mg precisely weighted were place in a Petri plate and afterwards irradiated by ultraviolet (UV) light for 1 h in order to sterilize them. A pellet of bacterial strain (*Escherichia coli* and *Enterococcus faecalis*, alternatively) was dissolved in 3 mL of autoclaved saline water in order to obtain a bacterial suspension of 105 colony-forming units (CFU)/mL.

After plating bacteria on medium agar, the samples were placed on the middle of the plate. *Escherichia coli* was incubated at 44 °C for 24 h, while *Enterococcus faecalis* was incubated at 36 °C for 48 h. The diameter of inhibition halos (IDs) in relation to Petri plate diameter (6 cm) were calculated. Three distinct measurements were carried out for each sample in order to determine the corresponding mean standard deviation.

## 3. Results and Discussion

First of all, the stoichiometry of these materials has been assessed by determining experimentally the Si content. Calculated and measured percentages of Si were summarized in Table 1, where good agreement between the two values for each material confirmed the assessed stoichiometry.

All the results obtained in this study are discussed in four distinct sections: the structural and morphological characterization (displayed in Section 3.1), the thermal behavior study (Section 3.2), the FTIR study (Section 3.3) and the analysis of antibacterial properties in pellets by treating the materials with *Escherichia coli* and *Enterococcus faecalis* (Section 3.4).

### 3.1. Results of the Structural and Morphological Analysis

The X-ray diffraction (XRD) spectra of the materials (in the form of fine powder) are reported in Figure 1.

It is evident that all the samples are amorphous and the XRD spectra of the materials show the same typical shape of the SiO_2_/PEG hybrid, which is practical superimposable with that of pure amorphous or semicrystalline SiO_2_, where a wide peak between 15 and 35° is observed, as displayed elsewhere [7].

Figure 2 displays the scanning electron microscopy (SEM) images of the synthesized hybrid materials. Regardless the addition of PEG, CGA, or both to silica matrix, the microstructure is not affected, and the samples are morphologically similar.

Altogether, it can be seen that homogeneous, very well reacted particles are formed. This fact is coherent with the aforementioned results of electron microanalysis using EDS displayed in Figure 3, while the supplementary material content in the materials is summarized in Table 1, where the amount of silicon observed by this method is in agreement with the assumed theoretical one, which indicates that the correct reaction occurred in the correct proportions.

### 3.2. FTIR Study

To better understand the interaction of silica with CGA and PEG, FTIR spectra (not shown) were recorded and the most significant bands are reported in Table 2. In all the spectra within the SiO_2_, the Si–O–Si asymmetric stretching and Si–O bending are observed at 1080 cm^−1^ and at 460 cm^−1^, respectively. The strong and broad signals between 3600 and 3300 cm**^−^**^1^ and that at 1640 cm**^−^**^1^, ascribed to stretching and bending –OH, respectively, suggest the presence of H– bonds between the inorganic and organic phases. Moreover, the presence of PEG and/or CGA in the synthesized hybrid materials, can be demonstrated by the presence of stretching and bending C–H that are not present in the spectrum of pure silica.

### 3.3. Thermal Analysis Investigation

The TG curves of the investigated materials are shown in Figure 4 along with that of pristine silica (S) for comparison purposes. In particular, the thermal behavior of the two CGA20- and PEG50-based materials (SC20, SP12C20, SP50C20 and SP50, SP50C10 and SP50C20 with increasing the PEG and CGA content, respectively) can be studied by observing Figure 4a,b, respectively.

Water, physically and chemically retained in the material with weak and moderately strong bonds, is released upon heating in such different close temperature ranges that it is not possible to discriminate between the two mass losses under the conventional experimental conditions. Figure 4a shows that up to 180 °C the higher the water content the lower the polymer content in the CGA20 wt%-based materials. Pure S has a lower amount of water than the PEG-free hybrid material (SC20), but higher water content than that of CGA20 wt%-based materials (Figure 4a, black line) and practically identical amount of SP50 (CGA-free material (Figure 4b, black and red lines). On the other hand, the first mass loss is practically superimposable for SP50C10 and SP50C20 (about 4% by mass), while remarkably higher water content is found in the same temperature range for CGA-free material (SP50). By comparing the plots, a and b of Figure 4 it is almost evident that a significant higher water amount is retained in the hybrid materials with a single organic component (SC20 and SP50). This larger amount of ‘entrapped’ water could be probably explained by the fact that more H-bonds can be formed between SC20 or SP50 and water (or residual alcohol) during the synthesis procedure with respect to the three component SPC hybrids.

In the presence of three component hybrid materials (namely, S + P + C, SP12C20, SP50C10 and SP50C20) the formation of more H-bonds between the inorganic S and the two organic P and C moieties makes less available those needed to retain water (or residual alcohol) molecules. As described elsewhere [46], pure S undergoes dehydroxylation, which is the characteristic slow release of water due to condensation of surface hydroxyl groups in the wide temperature range between 200 and 600 °C. The thermal degradation of both CGA and PEG occurs in the two component SC20 or SP50 hybrids practically in the same temperature range (Figure 4a,b). The significant difference in the amount of PEG contained either in SP12C20 and SP50C20 do not affect the temperature of the almost superimposable thermal degradation of CGA and PEG, while the mass loss percentages is strictly dependent on the PEG content, as found in a previous study on PEG-rich hybrids (SP60 and SP70) [14].

### 3.4. Study on Bacterial growth

The antibacterial properties of the SiO_2_/PEG/CGA materials were investigated by analyzing the bacterial growth of *Escherichia coli* and *Enterococcus faecalis* on agar in the presence and the absence of the samples. Figure 5a shows the values of inhibition halo diameter when all the hybrids are placed in the presence of bacteria. It can be noticed that both bacterial strains are susceptible to samples and the halo diameter values increase with increasing the amount of CGA in the silica/PEG 50% hybrids.

In addition, we can observe a lower bacterial growth for *Enterococcus faecalis* compared to that with *Escherichia coli* (Figure 5b). This result is in accordance with those reported in our previous paper [61].

The effect on silica-based hybrids with 20 wt% of CGA without and with different amounts of the polymer (PEG-free or with 12 and 50 wt% of this component) is shown in Figure 6a,b. All the samples exhibit an excellent antibacterial activity against *Escherichia*
*coli* and *Enterococcus faecalis*, but no differences are noticed when the amount of polymer is increased in the hybrids. The antibacterial activity observed in these materials is due to the presence of CGA. Many studies are reported in literature about the molecular mechanism of this natural compound [44,45,46]. The damage of the bacterial cell membrane could be caused by the depletion of radical oxygen species (ROS) induced by the presence of CGA in the different materials [66]. Therefore, when the level of ROS decreases in the cells, numerous signaling pathways are affected allowing inhibition of bacterial growth [47].

We can deduce that PEG does not influence antibacterial properties of the hybrids. This result is also confirmed by observing the trend in plot b of Figure 6 where the percentage of bacterial growth in the presence of SiO_2_ seems to be comparable with that in SP50 (SiO_2_/PEG 50 wt%).

## 4. Conclusions

Organic–inorganic three-component hybrid materials with a well-defined composition have been prepared by the sol-gel method. The inorganic silica matrix reacted with two organic components, a polymer that increases the hydrophilicity of the resulted hybrids (in this study, polyethylene glycol) and a non-flavonoid polyphenolic compound, chlorogenic acid (CGA), because of its good antibacterial, anti-inflammatory, and antioxidant properties.

The XRD and SEM experiments confirmed that the SPC three-component hybrids were mainly amorphous and almost homogeneous, regardless of their composition (content of both PEG and ACG).

A larger amount of water was retained by the organic–inorganic two-component hybrids SP50 and SC20 with respect to either pure silica and SPC three-component hybrids, where the lower content of water was found in the PEG-rich materials (with 50 wt% of PEG), since very few H-bonds were available to bind water molecules.

To check the antibacterial properties of the hybrid materials, the bacterial growth was monitored by using two different bacterial strains: *Escherichia coli* a Gram negative and *Enterococcus faecalis* a Gram positive. All the samples exhibit an excellent antibacterial activity against *Escherichia coli* and *Enterococcus faecalis*, regardless of the amount of polymer in the hybrids. It can be deduced that PEG does not influence antibacterial properties of the hybrids. In particular, surprisingly, the percentage of bacterial growth in the presence of SiO_2_ seems to be comparable with that in the CGA-free material: SP50 (SiO_2_/PEG 50 wt%). The results obtained show also that despite the addition of PEG, used to improve mechanical properties of biomaterial, its presence does not negatively affect biological properties necessary for a good integration in human tissue. Indeed, the release of a bioactive compound (CGA) can avoid infection phenomena post implementation, preventing the early rejection of artificial implant. Finally, based on the on the results it can be concluded that the chosen procedure is able to create a material with great potential for future biomedical and pharmaceutical applications.

## Figures and Tables

**Figure 1 polymers-13-01586-f001:**
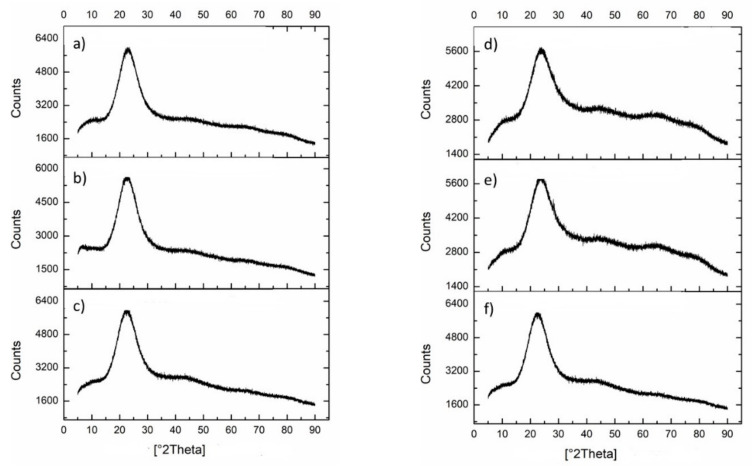
X-ray diffraction pattern of the fresh powder samples of the investigated materials: (**a**) SP50; (**b**) SP50C10; (**c**) SP50C20; (**d**) SC20; (**e**) SP12C20; (**f**) SP50C20 (equal to (**c**)). S = SiO_2_; P = PEG; C = ACG; 20 = 20 wt% and 50 = 50 wt%.

**Figure 2 polymers-13-01586-f002:**
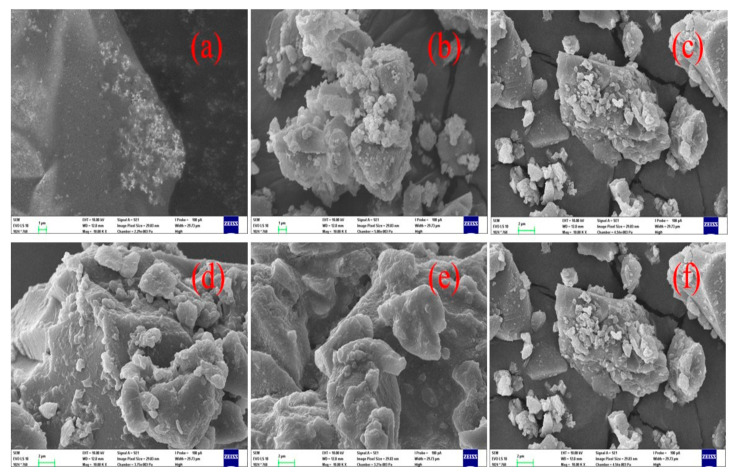
Scanning electron microscopy (SEM) images of the fresh powder samples of the investigated materials: (**a**) SP50; (**b**) SP50C10; (**c**) SP50C20; (**d**) SC20; (**e**) SP12C20; (**f**) SP50C20 (equal to (**c**)).

**Figure 3 polymers-13-01586-f003:**
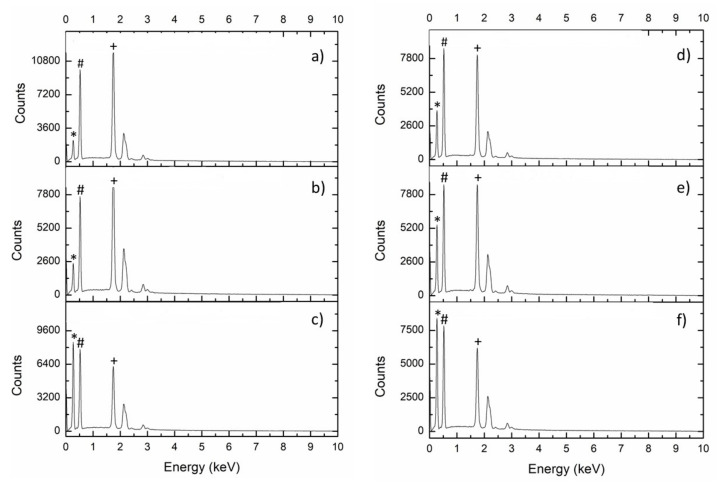
Energy-dispersive X-ray spectroscopy (EDS) spectra of the fresh powder samples of the investigated materials: (**a**) SP50; (**b**) SP50C10; (**c**) SP50C20; (**d**) SC20; (**e**) SP12C20; (**f**) SP50C20 (equal to (**c**)). Peaks related to: C (*), O (#) and Si (+).

**Figure 4 polymers-13-01586-f004:**
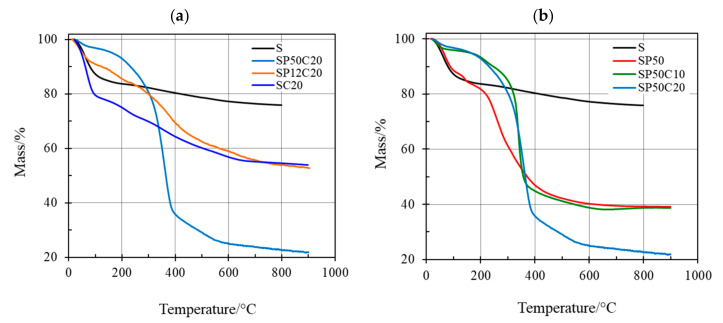
Thermogravimetric (TG) curves carried out at 10 °C/min under inert nitrogen atmosphere for (**a**) 20 wt% CGA-based hybrids; (**b**) 50 wt% PEG-based hybrids.

**Figure 5 polymers-13-01586-f005:**
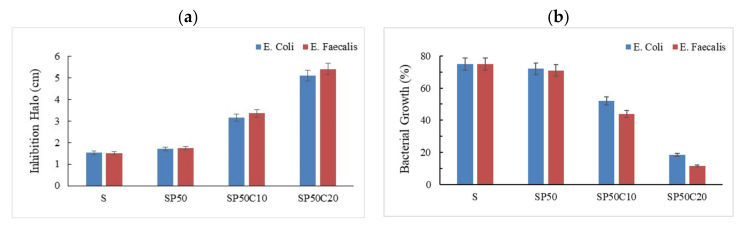
(**a**) Comparison of inhibition halo diameters produced by SiO_2_/PEG50% with different amount of entrapped CGA on *Escherichia coli* and *Enterococcus faecalis*. (**b**) Analysis of bacterial growth in the presence of silica/PEG/CGA hybrids. A negative control is represented by 100% confluency in culture dish without sample.

**Figure 6 polymers-13-01586-f006:**
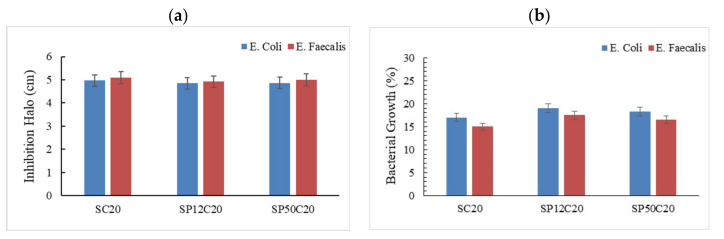
(**a**) Comparison of inhibition halo diameters produced by SiO_2_/PEG/CGA hybrids with the increase of polymer amount on *Escherichia coli* and *Enterococcus faecalis.* (**b**) Analysis of bacterial growth in the presence of silica/PEG/CGA hybrids. A negative control is represented by 100% confluency in culture dish without sample.

**Table 1 polymers-13-01586-t001:** List of materials. Calculated and measured percentages of Si in the investigated materials.

Sample (Symbol)	Materials	Calculated Si Content (wt%)	Measured Si Content (wt%)
a (SP50)	SiO2 + 50 wt% PEG 1	23.4	22.9
b (SP50C10)	SiO2 + 50 wt% PEG + 10 wt% CGA 2	18.7	20.5
c (SP50C20) 3	SiO2 + 50 wt% PEG + 20 wt% CGA	14.0	14.1
d (SC20)	SiO2 + 20 wt% CGA	37.4	37.3
e (SP12C20)	SiO2 + 12 wt% PEG + 20 wt% CGA	31.8	31.9

^1^ PEG = polyethylene glycol. ^2^ACG = chlorogenic acid. ^3^ Sample c is equal to sample f.

**Table 2 polymers-13-01586-t002:** Fourier transform infrared (FTIR) interpretation peaks of the synthesized material.

Sample	Common Position of Main Peaks (cm^−1^)					
S	3600–3300	-	1640	-	1080	470
P	3600–3300	2930, 2870	1640	1454	-	-
C	-	2930, 2870	1640	1454	-	-
SP50	3600–3000	2930, 2870	1640	1454	1080	470
SP50C10	3600–3000	2930, 2870	1640	1454	1080	470
SP50C20	3600–3000	2930, 2870	1640	1454	1080	470
SC20	3600–3330	2930, 2870	1640	1454	1080	470
SP12C20	3600–3000	2930, 2870	1640	1454	1080	470
Peak interpretation	–OH stretching	C–H stretching	–OH bending	C–H bending	Si–O–Si asymmetric stretching	Si–O bending

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
