# Peer review of "Synthesis, Structural, Morphological and Thermal Characterization of Five Different Silica-Polyethylene Glycol-Chlorogenic Acid Hybrid Materials"

_polymers, 2021, doi:10.3390/polym13101586_

Round 1
Reviewer 1 Report
The present study reports on study of the silica-polyethylene glycol-chlorogenic acid hybrid, and a series of experimental measurements have been carried out to support the claim of the thermal and antibacterial properties have been improved. It is interesting that the PEG and CGA have been introduced to the inorganic silica matrix. After carefully reading it, the following points are suggested to consider to improve this study.
- The synergistic effect of PEG and CGA has not been presented. Why these two polymers are used? How to determine the different roles of them to influence the properties of the composites?
2. The claim of “The TG analysis revealed a higher water content is adsorbed into the two component hybrids” lacks of support of experimental results. How about the effect of H-bond on the hybrid in terms of adsorbing water content.
3. Caption of Figure 3 is laid incorrectly. Furthermore, peaks in figures 1 and 3 should be marked for the readers.
4. “3.2. Results of the thermal analysis investigation” is suggested to revise to “3.2. Thermal analysis”; “3.3. Results of the bacterial growth investigation” is suggested to revise to “3.3. Study on bacterial growth”. In these sections, not only results has been presented, but also the discussion has also been presented. Therefore the “results” should be removed from the title.
5. In this study, the synergistic effect of two polymer matrix should be discussed, i.e., the glass transition temperature of the single polymer and hybrid are different from each other, and it is governed by the Gordon-Talyor theory. The following ref. can be cited to explain it, Haibao Lu and Wei Min Huang. On the origin of the Vogel-Fulcher-Tammann law in the thermo-responsive shape memory effect of amorphous polymers. Smart Materials & Structures. 2013, 22(10): 105021. On the other hand, why number of H-bond is increased, what is the working principle as the content of PEG or CGA increasing? Whether the hydrogen bonding strength has been improved, which enables the water absorption?
The effect of hydrogen bonding on the thermal and antibacterial properties should be discussed, the following refs. can be cited to explain it, (1) Panda, Pradeep Kumar; Yang, Jen-Ming; Chang, Yen-Hsiang. Water-induced shape memory behavior of poly (vinyl alcohol) and p-coumaric acid-modified water-soluble chitosan blended membrane. CARBOHYDRATE POLYMERS. 2021, 257, 117633. (2) Haibao Lu, Jinsong Leng and Shanyi Du. A Phenomenological Approach for the Chemo-Responsive Shape Memory Effect in Amorphous Polymers. Soft Matter. 2013, 9(14): 3851-3858.. In all, an useful study, I would like to recommend it after the revision.
Author Response
see the attached word file

Reviewer 2 Report
The manuscript titled “Synthesis, structural, morphological and thermal characterization of five different silica-polyethylene glycol-chlorogenic acid hybrid materials (Michelina Catauro, Pavel Siler, JiÅ™í Másilko, Roberta Risoluti, and Stefano Vecchio Ciprioti)” is very interesting and attractive to many researchers. The manuscript is clear and well structured. The research problem is well explained and supported by appropriate experiments. I recommend this manuscript for publication.
Author Response
see the attached word file

Reviewer 3 Report
The manuscript needs extensive English editing; e.g., to product at line 42, was growth at line 174.
Figure 3 is titled as XRF spectra. However, no details on the XRF measurements were given in experimental section.
In the introduction bulk references were given without adequate explanations. They have to be cut short and include only relevant ones.
What is special about this work compared to Ref. 12 and 58? It is not explained in introduction section.
FTIR studies should be added in the manuscript to support the hydrogen bonding theory proposed.
Overall the explanation of the results is weak. Abstract and conclusion can be made short with most important achievements from the study.
Author Response
see the attached word file

Round 2
Reviewer 1 Report
it is acceptable in the current version.
Reviewer 2 Report
I recommend the article for publication.
Reviewer 3 Report
Acceptable as present form